# New Indicator of Arterial Stiffness START—Is There a Prognostic Value of Its Dynamics in Patients with Coronary Artery Disease?

**DOI:** 10.3390/biomedicines12081638

**Published:** 2024-07-23

**Authors:** Alexey N. Sumin, Anna V. Shcheglova, Olga L. Barbarash

**Affiliations:** Federal State Budgetary Institution “Research Institute for Complex Issues of Cardiovascular Disease”, Blvd. Named Academician L.S. Barbarasha 6, 650002 Kemerovo, Russia; nura.karpovitch@yandex.ru (A.V.S.); olb61@mail.ru (O.L.B.)

**Keywords:** START arterial stiffness index, dynamics, coronary artery disease, coronary artery bypass grafting, long-term prognosis

## Abstract

The aim of the study was to evaluate the prognostic value of the one-year dynamics of the new index START in patients with coronary artery disease after coronary artery bypass grafting (CABG). Methods. Patients with coronary artery disease (n = 196) whose START index was assessed before CABG and one year after surgery. Depending on the dynamics of the stiffness index, three groups of patients were identified: 1st—with a decrease in haSTART (n = 79, 40.3%), 2nd—without dynamics (n = 52, 26.5%), and 3rd. Patients were followed for 10 years, and groups were compared for all-cause death, myocardial infarction, stroke/transient ischemic attack, and a composite endpoint. Results. In the group with an increase in the haSTART index, type D personality was identified more often (53.8%) than in the group without changes in haSTAR (26.9%) or with a decrease in the haSTAR index (34.2%) (*p* = 0.008). In the long-term follow-up period, death from all causes was significantly more common in the group with an increase in haSTART (33.9%) and in the group without changes in haSTART (23.1%) than in the group with a decrease in haSTART (11.4%, *p* = 0.005). Patients with an increase in haSTART more often had MACE (death, MI, stroke/TIA)—in 47.7% of cases (*p* = 0.01), compared with patients with a decrease in haSTART (in 24.1% of cases) and without change in haSTART (by 30.8%). Kaplan–Meier curves revealed better long-term survival rates in the group with a decrease in the haSTART index (*p* = 0.024). Multivariate analysis showed that a decrease in the haSTART index one year after CABG was associated with a decrease in mortality (HR 0.462; 95% CI 0.210–1.016; *p* = 0.055). Conclusions. The dynamics of the haSTART arterial stiffness index one year after CABG has prognostic significance in the long-term follow-up period. In addition, in the group with an increase in the haSTART index, personality type D is more common. Further studies need to study which interventions in patients with coronary artery disease can cause favorable dynamics in the haSTART index and to what extent psychological characteristics can influence these dynamics.

## 1. Introduction

The assessment of arterial stiffness is a convenient integral indicator that summarizes the influence of various risk factors and therapeutic effects on the arterial wall and, as a result, has an association with prognosis both in epidemiological population studies and in patients with cardiovascular diseases. It is not surprising that expert recommendations [1,2] suggest assessing this parameter in various categories of patients. However, in routine clinical practice, the assessment of arterial stiffness is still not widely used; it seems that these indicators remain a subject of interest only to different groups of researchers. Apparently, there are several reasons for this situation. On the one hand, the most common indicator, pulse wave velocity (PWV), has several disadvantages (the need for qualified personnel, technical difficulties in registration, dependence on blood pressure level), which complicates the possibility of dynamic assessment [1]. Another recently popular indicator, the cardio-ankle vascular index (CAVI), does not depend on blood pressure levels, which makes its dynamic assessment possible [3]. Currently, the association of the CAVI index with prognosis has been shown in population studies [4] and in patients with cardiovascular diseases [5]. However, its evaluation has been carried out primarily in Asian populations, and the ability to evaluate it in other populations is still limited. Since the CAVI index is determined in a non-standard section of the arterial bed (in contrast to methods for assessing pulse wave velocity) [6], attempts to improve it continue. For example, the CAVI0 index has been proposed [7,8]. In addition, Russian researchers have proposed a new indicator of arterial stiffness—the START index (STiffness ARTerial), as well as the CAVI index, leveling the effect of blood pressure, only using a formula based on other physical principles [9]. Unlike the classical stiffness parameter β, the formula for calculating the new index uses the law of conservation of mass and momentum, a standard method for obtaining conditions at a discontinuity, where the pulse wavefront is modeled as a discontinuity and takes into account nonlinear effects affecting the speed of a large amplitude wave [9].

Published studies have shown that this index correlated with the CAVI indicator both in healthy individuals [10] and in various categories of patients [11], although its severity varies depending on gender and age differed from those changes in the CAVI index [9]. However, the question remains open—to what extent this indicator can have a prognostic value similar to those already identified, say, for CAVI.

It should be noted that the possibility of dynamic assessment of arterial stiffness allows one to obtain additional information about the clinical condition and prognosis of patients. For example, for the CAVI index, it was shown that its increase after six months/year was associated with an unfavorable prognosis in patients who underwent acute coronary syndrome [12] or coronary artery bypass surgery [13]. In addition, Japanese studies showed an association between an increase in CAVI and severe psycho-emotional stress and with the subsequent development of cardiovascular events [14,15]. The question arises about the possible influence of the personal characteristics of patients on these associations [16]. In this regard, it is appropriate to remember that a predisposition to psychological distress (or, in other words, the presence of type D personality) is associated with an unfavorable prognosis in patients with coronary artery disease [17,18]. Therefore, it would also be interesting to evaluate whether personality type D affects the dynamics of the START index in patients with coronary artery disease.

This was the basis for conducting this study, the purpose of which was to evaluate the prognostic value of the one-year dynamics of the new index START in patients with coronary artery disease after coronary artery bypass grafting. 

## 2. Methods

### 2.1. Study Population

The single-center study included 479 patients with coronary artery disease aged 35 to 80 years, median age of 62.0 [61.0; 62.0] years, aimed at coronary artery bypass grafting (CABG) in the period from 2012 to 2013 at the Research Institute of Complex Problems of Cardiovascular Diseases, Kemerovo (Figure 1). Criteria for inclusion in the study: patients with coronary artery disease, informed consent of the patient to conduct the study. Non-inclusion criteria: ankle-brachial index (ABI) value ≤ 0.9 because the presence of hemodynamically significant stenosis of the arteries of the lower extremities distorts the true indicators of arterial stiffness (Figure 2). This study was performed in accordance with the principles of the Declaration of Helsinki, and the study protocol was approved by the local ethics committee of the Research Institute for Complex Problems of Cardiovascular Diseases (protocol No. 20110216). All patients signed informed consent before inclusion in the study.

### 2.2. Data Collection

Baseline patient data were obtained from the institute’s electronic registry database. For each patient, the following indicators were collected: anamnestic data, risk factors, comorbid conditions, severity of clinical manifestations of coronary heart disease, and laboratory and instrumental examination data before coronary artery bypass surgery. The condition of peripheral arteries was assessed in patients using duplex ultrasound. Arterial stiffness was assessed using CAVI, and the START index was calculated based on data obtained from volumetric sphygmography. We also assessed perioperative findings of the coronary artery bypass grafting and susceptibility to psychological distress (by identifying type D personality). All patients received drug therapy determined by the attending physician, taking into account the recommendations.

### 2.3. CAVI Measurement

Peripheral artery stiffness was assessed by the CAVI method using a VaSera VS-1000 sphygmomanometer (Fukuda Denshi, Tokyo, Japan) according to a previously described protocol [13]. During the study, the patient was dressed in light clothing that did not compress the body (socks, stockings, tights, etc.) or a special robe for research. The measurement was carried out in a quiet room in the first half of the day, two hours after eating. The temperature in the room where the study was carried out was about 25 °C. If the patient had any physical activity before the study, then a 20-minute rest was required; if there was no physical activity beforehand, then a 5-minute rest before the start of the study was sufficient. During the examination, the patient lies face up on a wide couch. The assessment of the CAVI index is automated and, therefore, does not depend on the operator’s qualifications and has high reproducibility [19,20]. The automatic calculation of the right and left index (R-CAVI and L-CAVI) is based on the stiffness parameter β. Since this parameter does not depend on blood pressure, the higher β, the lower the compliance and the greater the stiffness of the arteries. The device also assessed ABI when measuring blood pressure in the upper and lower extremities.

### 2.4. START Measurement

The basis for determining the START index was the data obtained when assessing arterial stiffness using the VaSera VS-1000 Fukuda Denshi device (Tokyo, Japan). The sphygmograph automatically calculated the pulse wave velocity (haPWV) between the heart and the left and right ankle, measured using a PCG phonocardiogram (II tone) and a plethysmogram obtained using smears attached to four limbs [3]. From sphygmography reports using an online calculator (https://stelari-start.com/), a new START stiffness index was calculated [9]. Taking into account the approach of the developers, the new index is designated according to the measured vascular region, i.e., in this case, as “haSTART”. We presented a detailed description of the calculation of the START index earlier [11]. This method of assessing stiffness is based on research by I.B. Bakholdin, described in previous publications [21]. The new rigidity parameter obtained by this researcher is based on the law of conservation of mass and momentum. This is its difference from the β index. As the author of the method stated, the new stiffness parameter better describes the state of elastic arterial walls with pronounced differences between systolic (SBP) and diastolic blood pressure (DBP) [9].

### 2.5. Type D Personality

To determine personality type D, we used the DS-14 questionnaire, which includes subscales NA (“negative affectivity”) and SI (“social inhibition”), validated in Russian by G. Pushkarev et al. [22]. The adequacy of the internal structure of the Russian version of DS14 is confirmed by the fact that Cronbach’s alpha for NA was 0.78 for SI—0.74. The questionnaire contains 14 questions with the following answer options: incorrect, rather incorrect, difficult to say, possibly true, and absolutely true. Each answer has its own score; if there are 10 points or more on each of the NA and SI scales, personality type D is established.

### 2.6. Laboratory Methods

In a blood sample taken on an empty stomach, the following indicators were assessed: the level of glucose, creatinine, total cholesterol (TC), and low-density cholesterol (LDL-C). The glomerular filtration rate was calculated using the CKD-EPI formula. Laboratory methods were strictly standardized and performed on a Konelab I 30 apparatus (Thermo Fisher Scientific Oy, Vantaa, Finland) using the same sets of reagents in clinical laboratories. Lipid spectrum indicators were determined by enzymatic colorimetric methods.

### 2.7. Follow-Up Measures One Year

After one year, patients were re-examined at the study center. At the same time, repeated measurement of arterial stiffness was performed in 341 patients. Depending on the dynamics of the stiffness index, three groups of patients were divided: 1st group—a decrease in the haSTART index (decrease by 1 unit or more), 2nd group—the haSTART index without changes (or its slight dynamics), and 3rd group—increase in the haSTART hardness index (increase by 1 unit or more). During the examination of the patient, clinical and anamnestic data and laboratory and instrumental parameters were also assessed.

### 2.8. Follow-Up Measures for 10 Years

Information about the long-term prognosis of patients was collected through telephone contacts with patients and analysis of the regional medical information system. This took into account both death from all causes and the development of cardiovascular events (death, non-fatal myocardial infarction, and stroke), i.e., MACE. Additionally, the therapy received in the groups at the time of contact with patients was assessed. 

### 2.9. Statistical Analyses

For statistical processing, the programs “STATISTICA 8.0” (Dell Software, Inc., Round Rock, TX, USA) and the SPSS 17.0 software package were used. The distribution of quantitative data was checked using the Shapiro–Wilk test. Considering that the distribution of all quantitative characteristics differed from normal, they are presented as median, upper, and lower quartiles (Me [Q 25; 75]). To compare the 3 groups, the Kruskal–Wallace test was used, with a further pairwise comparison of groups using the Mann–Whitney test. Qualitative data were assessed using the χ^2^ test. When the number of observations was small, Fisher’s exact test with Yates’ correction was used. Bonferroni correction was used to address the problem of multiple comparisons. To compare survival during follow-up, the Kaplan–Meier survival method was used using the log-rank test (long-term survival and long-term event-free survival were assessed). At the same time, patients with a decrease in the haSTART index during the year were compared with patients with no dynamics/increase in this index. Risk factors for death from all causes or a composite endpoint were assessed using Cox proportional hazards in the same groups. Differences were considered statistically significant at *p* < 0.05.

## 3. Results

The study flowchart is detailed in Figure 1. The haSTART index was re-evaluated one year after CABG in 341 patients. Long-term results were studied in 196 patients (58% of initially included patients), on average 9.7 ± 0.9 years after CABG surgery. Additional analysis (Appendix A) did not reveal differences at the initial examination in the groups with and without information in the long-term period. Depending on the annual dynamics of the haSTART index, 3 groups of patients were formed: 1st—with a decrease in haSTART (n = 79, 40.3%), 2nd—with no dynamics (n = 52, 26.5%), and 3rd—with an increase in the haSTART index (n = 65, 33.2%).

Table 1 shows the initial preoperative data of patients with coronary artery disease. Men dominated both groups. Growth was statistically significantly lower in patients with a decrease in the haSTART index compared with the group without a change in haSTART (*p* = 0.002). Differences were identified in the frequency of presence of patients with type D—in the group with an increase in the haSTART index. It was detected statistically significantly more often (53.8% of cases) than in the group without changes (26.9% of cases) or with a decrease in the index (34.2% of cases) (*p* = 0.008). There were no statistically significant differences in clinical parameters and laboratory and instrumental indicators between the groups, either initially or after one year (Table 2).

Indicators of hemodynamics and vascular stiffness according to sphygmography data initially and one year after CABG are presented in Table 3. A year later, an increase in the level of SBP and DBP was detected compared to the initial indicators. Statistically significantly higher values of haPWV and CAVI were noted in the group with an increase in haSTART, in comparison with the group where the index decreased or did not change over the course of the year (*p* < 0.001).

Patients in the analyzed groups did not have statistically significant differences in coronary lesions (Figure 3). The main parameters of the intraoperative period also did not differ between groups (Table 4).

When analyzing the use of drug therapy in the prehospital stage, the group with an increase in the haSTART index was statistically significantly less likely to take calcium channel antagonists than the comparison groups (*p* = 0.037). During one year of observation, there was a trend towards a higher frequency of taking drug therapy, but without intergroup differences (*p* > 0.05). Over the entire follow-up period, there was a trend towards a higher frequency of β-blockers (*p* = 0.0033), statins (*p* = 0.021), and aspirin (*p* = 0.0032) in the group with a decrease in haSTART than in the group without a change in haSTART or the group with increasing haSTART (Figure 4).

In the long-term follow-up period, death from all causes was significantly more common in the group with an increase in haSTART (33.9%) and in the group without changes in haSTART (23.1%) than in the group with a decrease in haSTART (11.4%, *p* = 0.005). Patients with an increase in haSTART were more likely to have MACE (death, MI, stroke/TIA)—in 47.7% of cases (*p* = 0.01), compared with patients with a decrease in haSTART (in 24.1% of cases) and without changes in haSTART (in 30.8%) (Table 5).

Kaplan–Meier curves revealed a better long-term prognosis in the group with a decrease in the haSTART index compared with the group with no change/increase in the haSTART index (Figure 5 and Figure 6). The differences were statistically significant for survival rates (Figure 5, Appendix A; *p* = 0.024, *p* = 0.054, and *p* = 0.044, respectively, for log-rank, Breslow, and Taron-Ware tests). However, event-free survival did not differ between groups (Figure 4, Appendix A; *p* = 0.133, *p* = 0.278 and *p* = 0.227, respectively, for log-rank, Breslow and Taron-Ware tests).

Table 6 shows factors associated with all-cause mortality according to Cox analysis. According to univariate analysis, smoking one year after CABG and the presence of class 3 CHF before surgery were significantly associated with an increase in mortality, and a decrease in the haSTART index one year after CABG was associated with a decrease in mortality. Multivariate analysis showed that a decrease in the haSTART index one year after CABG was less associated with a decrease in mortality (hazard ratio 0.462; 95% confidence interval 0.210–1.016; *p* = 0.055). On the contrary, smoking one year after CABG (risk ratio 2.451; 95% confidence interval 1.126–5.336; *p* = 0.024), the presence of CHF 3 FC (risk ratio 1.645; 95% confidence interval 1.156–2.341; *p* = 0.006), as well as the number shunts during CABG surgery (hazard ratio 1.598; 95% confidence interval 1.018–2.507; *p* = 0.042) were associated with increased all-cause mortality.

Table 7 shows factors associated with the development of the combined endpoint according to Cox analysis. According to the results of univariate analysis, smoking one year after CABG (hazard ratio 4.241; 95% confidence interval 1.718–10.471; *p* = 0.002), carotid artery stenosis before CABG (hazard ratio 2.130; 95% confidence interval 1.071–4.237; *p* = 0.031), the presence of angina one year after CABG (hazard ratio 2.543; 95% confidence interval 1.008–6.416; *p* = 0.048), as well as index haSTART-L before CABG was associated with the development of the combined endpoint. The likelihood of its development was reduced by a decrease in the haSTART index one year after CABG (risk ratio 0.393; 95% confidence interval 0.187–0.826; *p* = 0.014) and increasing age (risk ratio 0.936; 95% confidence interval 0.892–0.982; *p* = 0.007). However, the results of multivariate analysis did not identify independent factors associated with the development of the combined endpoint.

## 4. Discussion

In the present study, we showed for the first time that serial assessment of the new haSTART arterial stiffness index has prognostic value in long-term follow-up of CAD patients after CABG surgery. First, in the group with an increase in the haSTART index, death from all causes and the development of a combined endpoint were more often observed during follow-up. Second, the group with a decrease in the haSTART index one year after CABG had better survival compared to patients with no dynamics/increase in this index. In addition, a decrease in the haSTART index was associated in univariate Cox analysis with both improved survival and less frequent development of the combined endpoint in patients. Thirdly, an additional assessment of personality type D revealed that in the group with an increase in the haSTART index a year later, this personality type was more common.

To date, no studies have been conducted on the prognostic value of the START arterial stiffness index. This index was considered by its developers as an alternative to the CAVI index since it also does not depend on blood pressure levels. Previous studies have compared these indices. Thus, in a multicenter study based on five clinical centers with the inclusion of 928 people—healthy individuals and patients with arterial hypertension—it was shown that the haSTART arterial stiffness index significantly correlates with the CAVI index, without demonstrating significant differences in the quantitative relationship with blood pressure, BMI, heart rate and gender in various subgroups of subjects, i.e., despite the differences in the methods for calculating indices, a comparative analysis showed that the indices are numerically interrelated, and also behave similarly with regard to correlations with various demographic and physiological indicators [23]. Subsequently, similar correlations between the haSTART and CAVI indices were identified in patients with coronary artery disease [11]. It is important to emphasize that the haSTART index follows the same patterns as the CAVI, which suggests that the indices are similar in assessing arterial stiffness and, therefore, suggest similar effectiveness of haSTART as a prognostic marker for assessing cardiovascular risk. However, this assumption requires additional analysis of the prognostic value of the haSTART index. An example can be given of a modification of the CAVI index—CAVI0, which was developed with the aim of further improving this index and was intended to level out the dependence on blood pressure [6]. However, CAVI has demonstrated greater accuracy in clinical situations and has better prognostic value compared with both CAVI0 and another measure of arterial stiffness, brachial-ankle PWV [7]. In the present study, we were able to demonstrate the prognostic value of the dynamics of the haSTART index, confirming the assumption of the authors of the above-mentioned article [23].

Another difference in this study is the serial assessment of the haSTART index. This approach allows for a comprehensive assessment of the effect of rehabilitation and drug interventions in various categories of patients. Judging by our data, it was the dynamics of the haSTART indicator during the year that turned out to be more significant in predicting death from all causes compared to the initial values of this index. This is quite consistent with previously published studies on serial assessment of the CAVI index. It was previously shown that unfavorable annual dynamics of CAVI in patients with dyslipidemia and risk factors were associated with the development of MACE during follow-up for 5 years [24]. Similar results were obtained in patients with newly diagnosed coronary artery disease—a persistent increase in CAVI over 6 months was a predictor of the development of MACE over the next three years [12]. In a study by our scientific group, an analysis of the dynamics of the index showed that the persistence of pathological CAVI over a year was accompanied by worsening survival and event-free survival during long-term follow-up [13].

At first view, the more frequent identification of patients with type D personality in the group with increasing haSTART was surprising. It would seem that why exactly this indicator influenced the dynamics of this index but not the generally accepted known risk factors (arterial hypertension, smoking, diabetes mellitus, BMI), which did not differ between the groups. This contradiction is perhaps explained by studies in Japan on the dynamic assessment of the CAVI index [14,15]. It was shown that an increase in CAVI in response to severe stress (earthquake) was accompanied by an increase in the number of cardiovascular events in patients with coronary artery disease [14]. In addition, in healthy individuals with increasing levels of biomarkers of chronic psychological stress, an increase in the CAVI index was detected [25]. Even a relatively short 5-minute mental test (mathematical calculation) was accompanied by a significant increase in arterial stiffness within 30 min [26]. These observations help explain the more frequent identification of personality type D in the group of patients with an increase in the haSTART index. Indeed, individuals with this personality type are prone to more pronounced and protracted stress reactions, as has been previously shown [27]. It is also interesting that such an association of type D with a tendency to unfavorable dynamics of the haSTAR index may serve as a manifestation of one of the mechanisms of association of type D with an unfavorable prognosis [28]. It can be assumed that increased arterial stiffness is a factor mediating the effect of psycho-emotional stress on prognosis. However, in the present study, this assumption has not yet been confirmed—we have not identified the influence of personality type D on the prognosis in the cohort of patients we examined. Overall, the data from our study provide additional rationale for both the clinical use of the haSTART index and its serial evaluation.

Limitations of the study should be considered when considering its results. First, there are currently no normative values for the haSTART index, so we could not distribute patients into groups with normal or pathological haSTART values, as was the case in works on serial assessment of the CAVI index [12]. However, a simple assessment of the dynamics of the haSTART index (decrease/increase) allowed us to form groups of patients with coronary artery disease with different long-term prognoses. Second, we did not compare the prognostic value of the haSTART and CAVI indices; perhaps this could more accurately position the use of these indices in practice. This is a task for future research, although the haSTART index is being developed for those situations where the use of the CAVI index is not possible (for example, when using another device). Thirdly, the evaluation of the haSTART index was carried out retrospectively, which does not allow us to evaluate the effects of certain interventions aimed at correcting risk factors or medical therapy. In addition, the retrospective design of the study did not allow us to assess the accuracy and reproducibility of the START index with repeated measurements. In future prospective studies, we plan to examine the accuracy and reproducibility of the START index with repeated measurements. Fourth, the patient cohort is limited to patients with CAD who underwent CABG surgery. Whether the results of this study can be generalized to other diseases remains unclear. It should also be noted that although a significant effect of a decrease in the START index over a year on survival was obtained in intergroup comparisons, log-rank analysis, and univariate Cox analysis, they do not were confirmed by multivariate Cox analysis (*p* = 0.055). Fifthly, the study was conducted in one of the Russian centers; it is unclear whether its results can be extended to other centers and regions. Accordingly, the data from this study require confirmation in a multicenter study. However, the results of the present study can be used to justify such a multicenter study. Nevertheless, the very fact of identifying the prognostic impact of the haSTART index deserves attention and confirmation in further studies.

## 5. Conclusions

The present study showed that the dynamics of the haSTART arterial stiffness index one year after CABG has prognostic value in long-term follow-up of patients with coronary artery disease. In addition, in the group with an increase in the haSTART index, personality type D is more common. Further studies need to study what interventions in patients with coronary artery disease can cause favorable dynamics in the haSTART index and to what extent psychological characteristics can influence these dynamics.

## Figures and Tables

**Figure 1 biomedicines-12-01638-f001:**
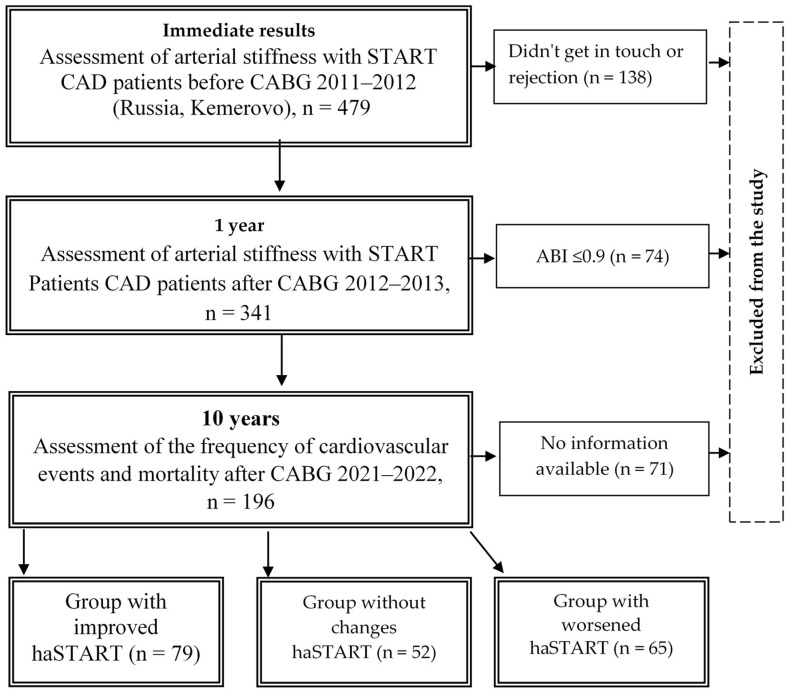
Flow chart of the study design from screening to completion of the trial. Notes: START—stiffness off arterial; CABG—coronary artery bypass grafting; ABI—ankle-brachial index; CAD—coronary artery disease.

**Figure 2 biomedicines-12-01638-f002:**
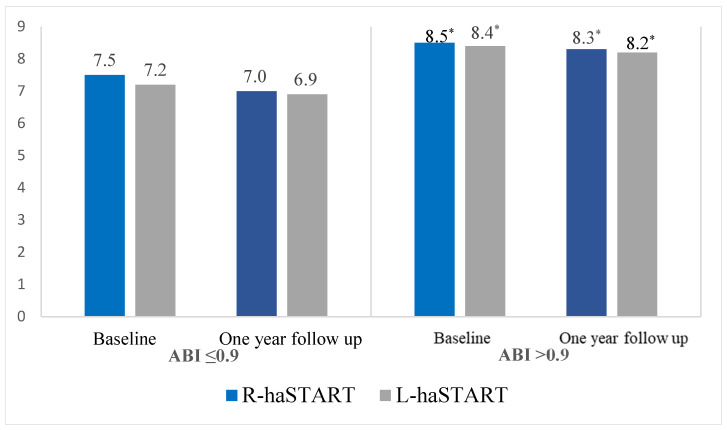
Comparison of the new arterial stiffness index START depending on the ABI value. Notes: * *p* < 0.001—significant differences in pairwise comparison of groups ABI ≤ 0.9 and ABI > 0.9; START—arterial stiffness index; ABI—ankle-brachial index; R—right; L—left.

**Figure 3 biomedicines-12-01638-f003:**
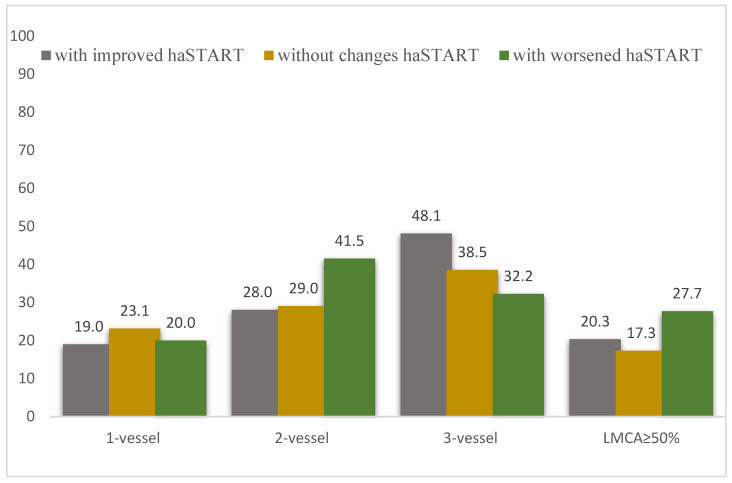
The severity of coronary artery damage in groups of patients with coronary artery disease, depending on the dynamics of the haSTART index at baseline and one year after CABG. Notes: LMCA—left main coronary artery; *p* > 0.05.

**Figure 4 biomedicines-12-01638-f004:**
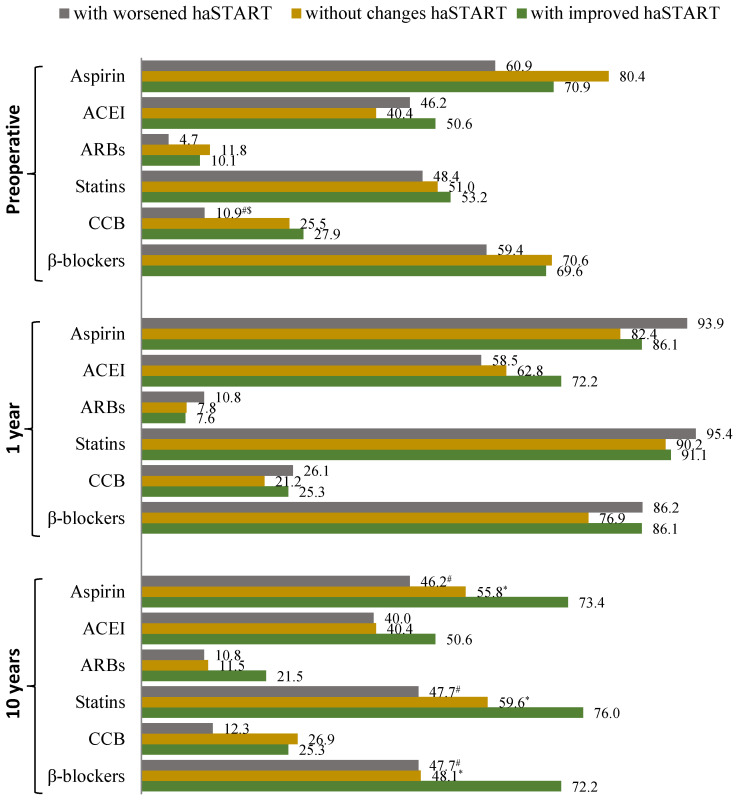
Medical treatment by patients in groups of patients depending on the dynamics of the haSTART after CABG. Notes: ARBs—Angiotensin II receptor blockers, ACEI—Angiotensin-converting-enzyme inhibitor, CCB—calcium channel blockers; *—significant differences in pairwise comparison of groups with improved haSTART and without changes haSTART; ^#^—significant differences in pairwise comparison of groups with improved haSTART and with worsened haSTART; ^$^—significant differences in pairwise comparison of groups without changes haSTART with worsened haSTART.

**Figure 5 biomedicines-12-01638-f005:**
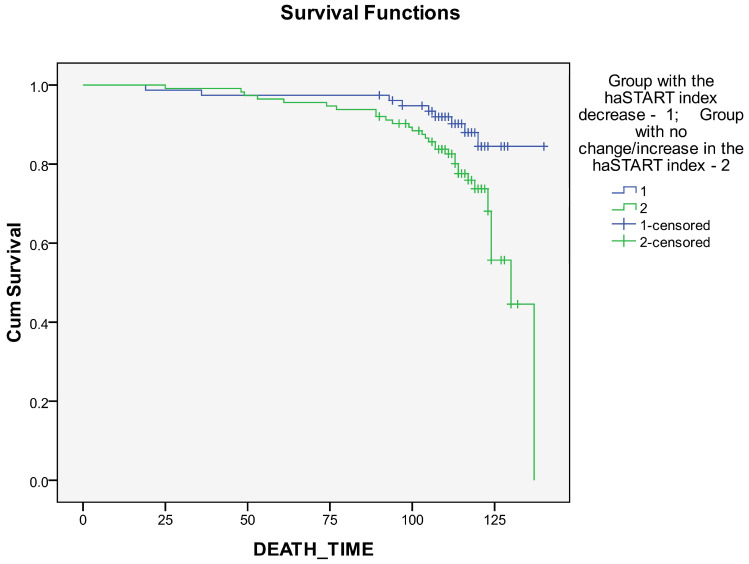
Impact of haSTART index dynamics within a year after coronary artery bypass grafting on long-term survival.

**Figure 6 biomedicines-12-01638-f006:**
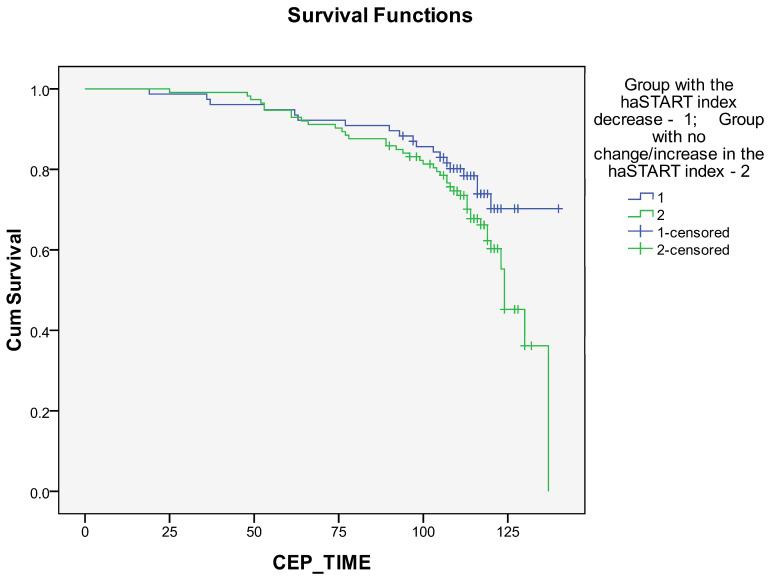
Impact of haSTART index dynamics within a year after coronary artery bypass grafting on long-term event-free survival.

**Table 1 biomedicines-12-01638-t001:** Initial clinical and demographic characteristics in groups of patients with coronary heart disease, depending on the dynamics of the haSTART index at baseline and a year after CABG.

Variables	Group 1 with Improved haSTART (n = 79)	Group 2 without Changes haSTART (n = 52)	Group 3 with Worsened haSTART (n = 65)	*p*-Value
Age (years)	58.0 [53.0; 64.0]	59.0 [56.0; 64.0]	61.0 [54.0; 64.0]	0.671
Male sex, n (%)	52 (65.8)	41 (78.9)	50 (76.9)	0.176
Height (cm)	168.0 [160.0; 175.0]	170.0 [166.0; 176.0] *	170.0 [163.5; 175.5]	0.026
Weight (kg)	80.0 [70.0; 90.0]	80.0 [73.5; 88.5]	79.0 [68.0; 85.0]	0.448
BMI (kg/m^2^)	29.3 [25.8; 32.0]	28.3 [25.9; 30.8]	27.9 [24.6; 31.3]	0.444
Smoking experience (years)	31.0 [20.0; 40.0]	30.0 [20.0; 40.0]	31.0 [24.0; 40.0]	0.742
Smoking, n (%)	19 (24.1)	11 (21.2)	20 (30.8)	0.459
Type D, n (%)	27 (34.2)	14 (26.9)	35 (53.8) ^#$^	0.008
Diabetes mellitus, n (%)	6 (7.6)	11 (21.2)	11 (16.9)	0.072
Arterial hypertension, n (%)	65 (82.3)	41 (78.9)	59 (90.8)	0.178
History TIA, n (%)	1 (1.3)	0 (0)	1 (1.5)	0.685
History stroke, (n%)	8 (10.1)	4 (7.7)	2 (3.1)	0.259
History MI, n (%)	45 (56.9)	33 (63.5)	38 (58.5)	0.752
Previous PCI, n (%)	4 (5.1)	3 (5.8)	4 (6.2)	0.959
Previous CABG, n (%)	1 (1.3)	0 (0)	1 (1.5)	0.685
Carotid endarterectomy, n (%)	1 (1.3)	0 (0)	2 (3.1)	0.391
Carotid artery stenosis ≥ 50%, n (%)	11 (21.2)	19 (24.1)	16 (24.6)	0.361

Notes: BMI—body mass index, TIA—trans-ischemic attack, MI—myocardial infarction, PCI—percutaneous coronary intervention, CABG—coronary artery bypass grafting abbreviations. *—significant differences in pairwise comparison of groups 1 and 2, ^#^—significant differences in pairwise comparison of groups 1 and 3, ^$^—significant differences in pairwise comparison of groups 2 and 3.

**Table 2 biomedicines-12-01638-t002:** Clinical and laboratory-instrumental data in groups of patients with coronary heart disease, depending on the dynamics of the haSTART index at baseline and one year after CABG.

Variables	Group 1 with Improved haSTART (n = 79)	Group 2 without Changes haSTART (n = 52)	Group 3 with Worsened haSTART (n = 65)	*p*-Value
Clinical assessment				
Angina baseline, n (%)	63 (79.8)	42 (80.8)	51 (78.5)	0.952
Angina after one year, n (%)	9 (11.4)	4 (7.7)	4 (6.2)	0.516
Heart failure NYHA II-III baseline, n (%)	20 (25.3)	15 (28.9)	19 (29.2)	0.846
Heart failure NYHA II-III after one year, n (%)	9 (11.4)	9 (17.3)	8 (12.3)	0.547
Laboratory findings				
Total cholesterol baseline, mmol/L	4.9 [4.2; 5.9]	4.6 [3.9; 5.5]	4.8 [4.0; 5.9]	0.446
Total cholesterol after one year, mmol/L	4.9 [4.0; 5.8]	4.9 [3.5; 5.8]	4.4 [3.9; 5.5]	0.521
LDL-C baseline, mmol/L	3.04 [2.2; 3.9]	2.9 [2.1; 3.4]	2.7 [2.1; 3.5]	0.479
LDL-C after one year, mmol/L	2.9 [2.3; 3.7]	2.8 [1.7; 3.8]	2.5 [2.0; 3.7]	0.533
Glucose baseline, mmol/L	5.5 [5.2; 6.4]	5.5 [5.0; 6.1]	5.6 [5.3; 6.5]	0.452
Glucose after one year mmol/L	5.5 [5.1; 6.4]	5.7 [5.3; 6.5]	6.0 [5.4; 7.1]	0.268
GFR CKD—EPI baseline, mL/min/1.73 m^2^	78.8 [65.6; 103.1]	83.3 [67.5; 100.1]	83.0 [62.2; 98.3]	0.861
GFR CKD—EPI after one year, mL/min/1.73 m^2^	92.8 [70.6; 111.3]	94.8 [65.5; 115.8]	86.6 [73.5; 104.4]	0.739
Echocardiography				
LV ejection fraction baseline, (%)	61.0 [51.0; 65.0]	60.0 [54.0; 63.0]	61.0 [54.0; 64.0]	0.740
LV ejection fraction after one year, (%)	62.0 [55.5; 65.0]	61.0 [50.0; 64.0]	61.0 [56.0; 65.0]	0.671
E/A baseline	0.8 [0.7; 1.2]	0.88 [0.7; 1.12]	0.8 [0.7; 1.12]	0.793
E/A after one year	0.71 [0.51; 1.2]	0.7 [0.5; 1.0]	0.68 [0.5; 1.0]	0.638

Notes: NYHA-New York Heart Association, LDL-C—low-density lipoprotein cholesterol, GFR—glomerular filtration rate, CKD-EPI-Chronic Kidney Disease Epidemiology Collaboration, LV—left ventricular, E/A—the ratio of the peak of the early to late transmitral flow.

**Table 3 biomedicines-12-01638-t003:** Sphygmography data in groups of patients with coronary heart disease, depending on the dynamics of the haSTART index at baseline and after one year of CABG.

Variables	Group 1 with Improved haSTART (n = 79)	Group 2 without Changes haSTART (n = 52)	Group 3 with Worsened haSTART (n = 65)	*p*-Value
SBP baseline, (mmHg)	129.0 [117.0; 145.0]	129.5 [121.0; 142.0]	130.0 [122.0; 139.0]	0.971
SBP after one year, (mmHg)	141.0 [127.0; 157.0]	142.0 [130.0; 153.0]	143.0 [131.0; 160.0]	0.653
DBP baseline, (mmHg)	81.0 [74.0; 87.0]	81.0 [73.5; 85.5]	79.0 [72.0; 85.0]	0.611
DBP after one year, (mmHg)	86.0 [80.0; 96.0]	89.0 [81.5; 96.0]	87.0 [81.0; 97.0]	0.891
R-PWV baseline, (m/s)	8.3 [7.6; 9.2]	7.9 [7.4; 8.5] *	7.7 [7.1; 8.,3] ^#^	0.003
R-PWV after one year, (m/s)	7.8 [7.2; 8.7]	8.2 [7.7; 8.9] *	8.8 [8.1; 9.5] ^#$^	<0.001
L-PWV baseline, (m/s)	8.1 [7.4; 9.1]	7.9 [7.3; 8.3] *	7.9 [7.0; 9.0] ^#^	0.024
L-PWV after one year, (m/s)	7.8 [7.2; 8.4]	8.1 [7.6; 8.8] *	8.9 [8.1; 9.6] ^#$^	<0.001
R-haSTART baseline	9.6 [7.9; 11.1]	8.25 [7.3; 9.4] *	8.0 [6.8; 9.1] ^#^	<0.001
R-haSTART after one year	7.6 [6.3; 8.6]	8.3 [7.5; 9.3] *	9.6 [8.3; 10.8] ^#$^	<0.001
L-haSTART baseline	9.1 [7.25; 10.5]	8.6 [7.1; 9.5]	7.8 [7.0; 9.0] ^#^	0.015
L-haSTART after one year	7.3 [6.3; 8.8]	8.0 [7.3; 9.2] *	9.6 [8.3; 11.3] ^#$^	<0.001
R-CAVI baseline	9.0 [8.0; 9.8]	8.3 [7.8; 9.0] *	8.1 [7.5; 8.8] ^#^	0.002
R-CAVI after one year	8.0 [7.4; 8.8]	8.4 [7.8; 9.1]	9.1 [8.5; 9.7] ^#$^	<0.001
L-CAVI baseline	9.0 [8.1; 9.7]	8.6 [7.9; 9.1] *	8.1 [7.7; 8.9] ^#^	0.001
L-CAVI after one year	7.9 [7.3; 8.8]	8.3 [7.7; 8.9]	9.1 [8.3; 9.6] ^#$^	<0.001
R-ABI p baseline	1.16 [1.08; 1.23]	1.14 [1.06; 1.24]	1.14 [1.04; 1.2]	0.396
R-ABI after one year	1.1 [1.01; 1.2]	1.09 [1.0; 1.18]	1.07 [0.98; 1.17]	0.975
L-ABI baseline	1.11 [1.03; 1.18]	1.09 [1.04; 1.14]	1.11 [1.02; 1.18]	0.868
L-ABI after one year	1.04 [0.94; 1.1]	1.05 [0.93; 1.12]	1.05 [0.95; 1.15]	0.801

Notes: SBP—systolic blood pressure DBP—diastolic blood pressure, ha—PWV—pulse wave velocity, START—stiffness off arterial; ABI—ankle-brachial index, CAVI—cardio-ankle vascular index, R-right; L-left. *—significant differences in pairwise comparison of groups 1 and 2, ^#^—significant differences in pairwise comparison of groups 1 and 3, ^$^—significant differences in pairwise comparison of groups 2 and 3.

**Table 4 biomedicines-12-01638-t004:** The main characteristics of coronary artery bypass surgery in groups of patients with coronary heart disease, depending on the dynamics of the haSTART index at baseline and one year after CABG.

Variables	Group 1 with Improved haSTART (n = 79)	Group 2 without Changes haSTART (n = 52)	Group 3 with Worsened haSTART (n = 65)	*p*-Value
EuroSCORE II, points	2.0 [2.0; 4.0]	2.0 [1.0; 3.0]	3.0 [1.0; 4.0]	0.761
EuroSCORE II, %	1.6 [0.9; 2.0]	1.5 [1.01; 2.3]	1.5 [0.9; 2.4]	0.659
Number of shunts	3.0 [2.0; 3.0]	2.0 [2.0; 3.0]	3.0 [2.0; 3.0]	0.346
Total operation time, minutes	246.0 [204.0; 273.0]	240.0 [204.0; 264.0]	240.0 [198.0; 300.0]	0.723
Cardiopulmonary bypass duration, minutes	98.0 [88.0; 114.5]	96.0 [77.0; 106.0]	93.5 [82.5; 109.0]	0.329
Cardiopulmonary bypass, n (%)	40 (76.9)	70 (88.6)	59 (90.8)	0.071
Ventriculoplasty, n (%)	1 (1.9)	5 (6.3)	2 (3.1)	0.405
Thrombectomy, n (%)	1 (1.9)	3 (3.8)	0 (0)	0.275
Carotid endarterectomy, n (%)	1 (1.9)	0 (0)	3 (4.6)	0.149
Radiofrequency ablation, n (%)	1 (1.)	1 (1.3)	2 (3.1)	0.744

Notes: EuroSCORE II—European Cardiovascular Risk Score.

**Table 5 biomedicines-12-01638-t005:** Complications of the ten-year period in groups of patients with coronary heart disease, depending on the dynamics of the START index initially and a year after CABG.

Variables	Group 1 with Improved haSTART (n = 79)	Group 2 without Changes haSTART (n = 52)	Group 3 with Worsened haSTART (n = 65)	*p*-Value
Total death, n (%)	9 (11.4)	12 (23.1) *	22 (33.9) ^#^	0.005
Cardiac death, n (%)	6 (7.6)	7 (13.5)	11 (16.9)	0.224
Non-fatal MI, n (%)	4 (5.1)	3 (5.6)	3 (4.6)	0.961
Non-fatal stroke/ TIA, n (%)	8 (10.1)	1 (1.9)	7 (10.8)	0.157
MACE, n (%)	19 (24.1)	16 (30.8)	31 (47.7) ^#$^	0.010
Resumption of angina, n (%)	35 (50.0)	19 (47.5)	17 (38.6)	0.485
PCI, n (%)	12 (15.2)	4 (7.7)	6 (9.2)	0.341
CABG, n (%)	0 (0)	0 (0)	1 (2.4)	0.271
Coronarography, n (%)	34 (43.0)	14 (26.9) *	16 (24.6) ^#^	0.037
Carotid endarterectomy, n (%)	5 (6.3)	3 (5.8)	2 (3.1)	0.655
Leg artery surgery, n (%)	2 (2.5)	2 (3.9)	2 (3.1)	0.912

Notes: TIA—trans-ischemic attack, MI—myocardial infarction, PCI—percutaneous coronary intervention, CABG—coronary artery bypass grafting abbreviations. *—significant differences in pairwise comparison of groups 1 and 2, ^#^—significant differences in pairwise comparison of groups 1 and 3, ^$^—significant differences in pairwise comparison of groups 2 and 3.

**Table 6 biomedicines-12-01638-t006:** Risk factors for mortality of patients after CABG during 10-year follow-up (Cox proportional hazards).

Factor	Univariate Analysis (Enther Method)			Multivariate Analysis (Forward Stepwise LR Method)		
	Hazard Ratio	95% CI	*p*-Value	Hazard Ratio	95% CI	*p*-Value
Sex (male)	0.709	0.208–2.419	0.583	-	-	-
Age	0.959	0.899–1.024	0.208	-	-	-
Body mass index	1.019	0.975–1.065	0.404	-	-	-
Total cholesterol	0.976	0.870–1.093	0.670	-	-	-
Arterial Hypertension	0.764	0.484–1.205	0.247	-	-	-
Diabetes Mellitus	0.621	0.232–1.657	0.341	-	-	-
Number of shunts	1.447	0.912–2.294	0.116	1.598	1.018–2.507	0.042
Angina baseline	2.584	0.867–7.705	0.089	-	-	-
Angina after one year	1.201	0.317–4.546	0.788	-	-	-
Smoking baseline	1.161	0.394–3.427	0.786	-	-	-
Smoking after one year	3.524	1.172–10.597	0.025	2.451	1.126–5.336	0.024
Index haSTART-R before CABG	1.155	0.895–1.491	0.268	-	-	-
Index haSTART-L before CABG	1.196	0.907–1.576	0.205	-	-	-
Decrease haSTART index one year after CABG	0.318	0.117–0.859	0.024	0.462	0.210–1.016	0.055
Heart failure NYHA II-III baseline	1.766	1.108–2.815	0.017	1.645	1.156–2.341	0.006
Carotid artery stenosis ≥ 50%	3.889	0.960–15.754	0.057	-	-	-

**Table 7 biomedicines-12-01638-t007:** Risk factors for the combined endpoint of patients after CABG during 10-year follow-up (Cox proportional hazards).

Factor	Univariate Analysis (Enther Method)			Multivariate Analysis (Forward Stepwise LR Method)		
	Hazard Ratio	95% CI	*p*-Value	Hazard Ratio	95% CI	*p*-Value
Sex (male)	1.696	0.733–3.928	0.217	-	-	-
Age	0.936	0.892–0.982	0.007	-	-	-
Body mass index	0.983	0.931–1.037	0.534	-	-	-
Arterial Hypertension	0.640	0.362–1.132	0.125	-	-	-
Total cholesterol	1.153	1.015–1.310	0.029	1.148	0.984–1.340	0.079
Diabetes Mellitus	0.682	0.320–1.457	0.323	-	-	-
Number of shunts	1.005	0.708–1.426	0.978	-	-	-
Angina baseline	1.924	0.902–4.100	0.090	-	-	-
Angina after one year	2.543	1.008–6.416	0.048	-	-	-
Smoking baseline	0.678	0.298–1.542	0.354	-	-	-
Smoking after one year	4.241	1.718–10.471	0.002	1.332	0.698–2.542	0.385
Index haSTART-R before CABG	1.140	0.951–1.367	0.156	-	-	-
Index haSTART-L before CABG	1.220	1.000–1.489	0.050	-	-	-
Decrease haSTART index one year after CABG	0.393	0.187–0.826	0.014	0.663	0.393–1.120	0.124
Heart failure NYHA II-III baseline	1.281	0.891–1.842	0.181	-	-	-
Carotid artery stenosis ≥ 50%	2.130	1.071–4.237	0.031	-	-	-

## Data Availability

The datasets used and/or analyzed during the current study are available from the corresponding author upon reasonable request.

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
