# Peer review of "New Indicator of Arterial Stiffness START—Is There a Prognostic Value of Its Dynamics in Patients with Coronary Artery Disease?"

_biomedicines, 2024, doi:10.3390/biomedicines12081638_

Round 1

Reviewer 1 Report

Comments and Suggestions for Authors

The manuscript presented by Sumin et al is aimed at validating an arterial stiffness index protocol. This protocol has limited use in the field of heart failure and has been developed in a very small environment, in which the authors of this work also participate. Due to this limited use, authors should first consider expanding their previous/local environment towards a multicentric study.

Regarding comparisons with the CAVI and PWV criteria, the values ​​are close enough not to modify previous databases based on them. Thus, the value of the START protocol requires additional assessment to ensure the potency vs the current measurements of vascular stiffness.

Perhaps the most novel aspect is the inclusion of the personality criterion, which aims to be a differentiating element.

In summary, the results presented have the limitations of being obtained from a single center in the Russian Federation, without being able to be validated in a multicenter manner; therefore, evaluation biases could affect the quality proposed for the START criterion.

Author Response

The manuscript presented by Sumin et al is aimed at validating an arterial stiffness index protocol. This protocol has limited use in the field of heart failure and has been developed in a very small environment, in which the authors of this work also participate. Due to this limited use, authors should first consider expanding their previous/local environment towards a multicentric study.

Response: We are grateful to the reviewer for useful comments that will help us improve the quality of our article. In response to your comment, I would like to clarify that we studied the new START vascular stiffness index in a cohort of patients with coronary heart disease, and not heart failure. In addition, before conducting a multicenter study, it is necessary to conduct studies in individual centers, so the present study can be considered as a pilot project in this direction. It is also worth mentioning that a multicenter study to study the clinical significance of the START index was conducted in centers of the Russian Federation (Vasyutin I.A. et al, ref. 23 in the article). However, it is much more difficult to conduct a multicenter study to study the prognostic value of the START index, which is why this pilot study was needed to justify the need for a multicenter study.

Regarding comparisons with the CAVI and PWV criteria, the values ​​are close enough not to modify previous databases based on them. Thus, the value of the START protocol requires additional assessment to ensure the potency vs the current measurements of vascular stiffness.

Response: We agree with you that the new START index is close in its values ​​to the existing CAVI and PWV indices. However, it allows one to overcome the existing shortcomings of existing indices. For PWV, this is the lack of standardization, dependence on operator qualifications and technical inconveniences of assessment. Unlike CAVI, the START index can be assessed in different areas of the vascular bed, and not be tied to a specific device model, which provides additional diagnostic capabilities. Therefore, we believe that the START index deserves study, including alongside current measurements of vascular stiffness.

Perhaps the most novel aspect is the inclusion of the personality criterion, which aims to be a differentiating element.

Response: We are grateful to the reviewer for highlighting this new scientific fact obtained in this study. We believe that the dependence of the increase in vascular stiffness on the personal characteristics of patients that we have identified will make it possible in the future to study the possibility of additional influence on the parameters of vascular stiffness (and, consequently, the prognosis) through behavioral interventions in this vulnerable group of patients.

In summary, the results presented have the limitations of being obtained from a single center in the Russian Federation, without being able to be validated in a multicenter manner; therefore, evaluation biases could affect the quality proposed for the START criterion.

Response: We agree that conducting this study in only one center limits the applicability of this index to other populations, and the data from this study require confirmation in a multicenter study. However, the results of the present study can be used to justify such a multicenter study. Taking into account the above, we have supplemented the section Limitations of the study.

Reviewer 2 Report

Comments and Suggestions for Authors

sophisticated statistical methodology would be needed for definitive conclusion. The following things are not criticism but points to increase the value of study and manuscript (please consider if possible):

1.      The comparative data of CAVI (or PWV) ‘for the long-term outcomes’ could be added; then, the relevance of both indicators on the outcomes could be discussed more. The comparison of AUC of each measurement for outcomes may be useful for it.

2.      The theoretical and logical explanation of using ‘one-year’ dynamics of START should be well described. Why is ‘one-year’ (unlike baseline, two-year or three-year)?

3.      As in Figure 1, the change of one-year START levels appeared to be few (even though there was a statistically significant) and many readers may think the relevance of the levels is minimal in the routine clinical practice. Explain more the relevance of the change in Discussion.

4.      The data of accuracy and reproducibility of START measured repeatedly should be more described.

5.      The measurement environments for START (i.e., room temperature, stress-relief, fasting-non-fasting, before and after CAVI or PWV measurement) could be detailed.

6.      The right- and left-sided data of CAVI and PWV presented in some Tables. Is that presentation by separated sides not often seen in prior studies. Why were the data of both-sides applied to this study?

7.      As in Table 1 and 2, how were the biochemical markers measured in the study? The information of assay methods and fasting/non-fasting conditions could be described more.

8.      The information on anti-diabetic drugs and anti-stress treatments could be included. This may be related to the results of type D personality.

9.      Most statisticians do recommend, as possible, using all variables, not only significant variables in the univariate analysis.

10.   As in Table 6 and 7, in terms of risk factors, body mass index, blood pressure, and cholesterol were included because these are established factors.

11.   As in Table 6 and 7, baseline and one-year START levels were assumed to be interrelated. So, the interaction for statistics could be fully carful. Rather one-year START levels only should be entered to the statistical model.

12.   As in Table 7, there were a lack of data by multivariate analysis. This seemed to be unnatural.

13.   Perhaps, the degree (level) of changes (i.e., -1, -2, -3…) in STARTS is also important for the outcomes. Such data may be added.

14.   In row 49-52, this sentence could have references.

15.   Based on Figure 1, there appeared to be many participants excluded. This might be a limitation to the conclusion.

16.   The expression of diabetes and diabetes mellitus was mixed.

Comments on the Quality of English Language

More editing of English language required.

Author Response

sophisticated statistical methodology would be needed for definitive conclusion. The following things are not criticism but points to increase the value of study and manuscript (please consider if possible):

Response: Dear reviewer, thank you for your careful reading of our article, your favorable opinion of it, and useful comments that helped us improve our manuscript.

  1. The comparative data of CAVI (or PWV) ‘for the long-term outcomes’ could be added; then, the relevance of both indicators on the outcomes could be discussed more. The comparison of AUC of each measurement for outcomes may be useful for it.

Response: We are grateful to the reviewer for this comment. In this article, we did not set ourselves the goal of comparing the START and CAVI  (or PWV) indices. We first of all wanted to study the prognostic significance of the dynamics of the START index. Since no such comparison was made, we note this in the Limitations of the Study section. Meanwhile, a comparison of the clinical significance of the START and CAVI indices has already been carried out previously, these results have been published (for example, in the article by Vasyutin et al, ref. 23 in the manuscript). Comparison of the prognostic value of these indices (as well as PWV) is planned in future studies.

  1. The theoretical and logical explanation of using ‘one-year’ dynamics of START should be well described. Why is ‘one-year’ (unlike baseline, two-year or three-year)?

Response: The assessment of the annual dynamics of the START index is explained by the design of the study, in which patients were examined in the clinic one year after coronary bypass surgery. This period after surgery turned out to be the most convenient for our patients. Also, in previously published studies, the dynamics of vascular stiffness indices (this concerned the CAVI index) were assessed after six months and a year. Apparently, this period was chosen empirically, since during this period of time one could expect the effect of preventive measures (physical activity, diet, medicine correction) on the condition of the vascular wall.

  1. As in Figure 1, the change of one-year START levels appeared to be few (even though there was a statistically significant) and many readers may think the relevance of the levels is minimal in the routine clinical practice. Explain more the relevance of the change in Discussion.

Response: Apparently there was some confusion. Figure 2 (it is clear that the reviewer is referring to this particular figure) shows the START index indicators at baseline and one year later in groups of patients with different initial ABI indexes. The differences in START index values ​​were statistically significant in these groups. Therefore, we did not include patients with ABI values ​​less than 0.9 in the study. Since START index was assessed in the general cohort, the dynamics were really insignificant. A different picture was observed when assessing changes in the START index in groups with different annual dynamics (Table 3). In the group with a decrease in the index, it decreased from 9.6 [7.9;11.1] to 7.6 [6.3;8.6], in the group with its increase, it increased from 8.0 [6.8;9.1] to 9.6 [8.3;10.8]. In our opinion, these changes cannot be called insignificant.

  1. The data of accuracy and reproducibility of START measured repeatedly should be more described.

Response: We are grateful to the reviewer for this comment. Unfortunately, the design of the study (the START index was calculated retrospectively in patients) did not allow us to assess its accuracy and reproducibility with repeated measurements. This is a limitation of the study; accordingly, we have supplemented the Limitations of the Study section. In future prospective studies, we plan to examine the accuracy and reproducibility of the START index with repeated measurements.

  1. The measurement environments for START (i.e., room temperature, stress-relief, fasting-non-fasting, before and after CAVI or PWV measurement) could be detailed.

Response: Thanks to the reviewer for this suggestion. We have supplemented the text of the manuscript with the following section:

"During the study, the patient was dressed in light clothing that did not compress the body (socks, stockings, tights, etc.) or a special robe for research. The measurement was carried out in a quiet room in the first half of the day, two hours after eating. The temperature in the room where the study was carried out was about 25°C. If the patient had any physical activity before the study, then a 20-minute rest was required; beforehand, then a 5-minute rest before the start of the study was sufficient. During the examination, the patient lies face up on a wide couch."

The assessment of the START stiffness index in our study was carried out based on the data obtained on the VaSera VS-1000 device retrospectively.

  1. The right- and left-sided data of CAVI and PWV presented in some Tables. Is that presentation by separated sides not often seen in prior studies. Why were the data of both-sides applied to this study?

Response: In previous investigations, we studied the prognostic value of the well-studied and proven CAVI index, which allowed us to use its average value from both sides. Considering that the START index is a new and little-studied parameter, we resorted to a more thorough analysis of the CAVI and PWV indicators (as well as the START index) on each side.

  1. As in Table 1 and 2, how were the biochemical markers measured in the study? The information of assay methods and fasting/non-fasting conditions could be described more.

Response: We have added the following text to the manuscript:

2.6. Laboratory methods

In a blood sample taken on an empty stomach, the following indicators were assessed: the level of glucose, creatinine, total cholesterol (TC), low-density cholesterol (LDL-C). The glomerular filtration rate was calculated using the CKD-EPI formula. Laboratory methods were strictly standardized and performed on a Konelab I 30 apparatus (Finland) using the same sets of reagents in clinical laboratories. Lipid spectrum indicators were determined by enzymatic colorimetric methods.

  1. The information on anti-diabetic drugs and anti-stress treatments could be included. This may be related to the results of type D personality.

Response: Patients with diabetes mellitus received appropriate oral antidiabetic drugs. Since there were no differences between groups with different annual dynamics of the START index in the frequency of concomitant diabetes mellitus, we did not study the frequency of taking antidiabetic drugs. Also in Tables 6 and 7, the presence of diabetes mellitus was not associated with the development of endpoints at 10-year follow-up.

In addition, in this study we did not study anti-stress methods in the treatment of patients with type D personality.

  1. Most statisticians do recommend, as possible, using all variables, not only significant variables in the univariate analysis.

Response: We agree with the reviewer that logistic regression analysis should include as many variables as possible. However, when carrying out calculations, we were repeatedly convinced that the inclusion of variables that did not have significant initial differences (in pairwise comparisons in groups) did not reveal significant differences in regression analysis. Therefore, we tried to include the most significant parameters in our model.

  1. As in Table 6 and 7, in terms of risk factors, body mass index, blood pressure, and cholesterol were included because these are established factors.

Response: Thanks for the suggestion, we have additionally included these indicators in the models, the results are presented in tables 6 and 7

  1. As in Table 6 and 7, baseline and one-year START levels were assumed to be interrelated. So, the interaction for statistics could be fully carful. Rather one-year START levels only should be entered to the statistical model.

Response: We tried to use for analysis, first of all, the dynamics of vascular stiffness indicators, by analogy with previous studies in this area (ref. 12-13 in the article). We also assessed the initial indicators of the START index, since in a number of previous studies it was on the basis of initial indicators that the clinical significance of stiffness indicators was assessed. Perhaps you are right and the index indicators in a year will be more informative; we will study this issue in future studies.

  1. As in Table 7, there were a lack of data by multivariate analysis. This seemed to be unnatural.

Response: Thanks for this suggestion, we have added multivariate analysis data to Table 7

  1. Perhaps, the degree (level) of changes (i.e., -1, -2, -3…) in STARTS is also important for the outcomes. Such data may be added.

Response: We agree with the reviewer that such additional analysis would be interesting. However, the relatively small sample size did not allow us to stratify patients according to the degree of dynamics of the START index

  1. In row 49-52, this sentence could have references.

Response: Unfortunately, in the text of the manuscript submitted to us for correction there is no line numbering and, accordingly, we cannot make changes according to the reviewer’s suggestions

  1. Based on Figure 1, there appeared to be many participants excluded. This might be a limitation to the conclusion.

Response: We agree with the reviewer that, due to the design features (the need to re-evaluate arterial stiffness in patients in a clinical setting, exclusion from the analysis of patients with an ABI less than 0.9, the length of follow-up), not all patients with an initial assessment of the SART index were included in the analysis. Perhaps this is another limitation of the study, although we would still consider it a design feature, the presence of exclusion criteria.

  1. The expression of diabetes and diabetes mellitus was mixed.

Response: We have made appropriate text corrections

Reviewer 3 Report

Comments and Suggestions for Authors

This is a report about the prognostic value of the one-year dynamics of the new index START in patients with coronary artery disease after coronary artery bypass grafting (CABG). Patients with an increase in haSTART more often had MACE in 47.7% of cases, compared with patients with a decrease in haSTART (24.1%) and without change in haSTART (30.8%). The result suggested that The dynamics of the haSTART arterial stiffness index one year after CABG has prognostic significance in the long-term follow-up period. There were several issues to be addressed.

# There was no data of multivariate analysis in Table 7. Even if there was no significant data, the result should be presented.

# How did you deal with patients who experience clinical events within one year after CABG?

# The value of LVEF seems to be relatively high in spite of high percentage of patients with MI. What was the reason for it? Did the study patients include patients with HFrEF?

Comments on the Quality of English Language

No comment

Author Response

This is a report about the prognostic value of the one-year dynamics of the new index START in patients with coronary artery disease after coronary artery bypass grafting (CABG). Patients with an increase in haSTART more often had MACE in 47.7% of cases, compared with patients with a decrease in haSTART (24.1%) and without change in haSTART (30.8%). The result suggested that The dynamics of the haSTART arterial stiffness index one year after CABG has prognostic significance in the long-term follow-up period. There were several issues to be addressed.

Response: Dear reviewer, thank you for your careful reading of our article, your favorable opinion of it, and useful comments that helped us improve our manuscript.

 # There was no data of multivariate analysis in Table 7. Even if there was no significant data, the result should be presented.

Response: Thanks for this suggestion, we have added multivariate analysis data to Table 7

 # How did you deal with patients who experience clinical events within one year after CABG?

Response: We may have been wrong, but we included nonfatal complications during the first year of follow-up in the total number of complications during the 10-year follow-up phase. It seemed to us that such an analysis was fully consistent with the main objective of the study (to study the association of the START index with long-term outcomes of CABG surgery). The exception was patients who died during the first year; naturally, we did not evaluate the START index after a year.

 # The value of LVEF seems to be relatively high in spite of high percentage of patients with MI. What was the reason for it? Did the study patients include patients with HFrEF?

Response: Indeed, in our cohort of patients, LVEF values ​​were relatively high, despite previous myocardial infarction. This is due to the fact that we did not include patients with low ejection fraction, valvular pathology and atrial fibrillation in the study.

Round 2

Reviewer 1 Report

Comments and Suggestions for Authors

The data are more suitable for a clinical journal

Author Response

The data are more suitable for a clinical journal

Response:

We thank the reviewer for this important observation. Indeed, our article is based on clinical material. However, the indicators studied (arterial stiffness and susceptibility to psychological distress) are difficult to study in experimental models. Therefore, it seems quite logical to study them in patients. It has been suggested that increased arterial stiffness is a factor mediating the effect of psychoemotional stress on prognosis. In addition, it is the initially increased arterial stiffness that contributes to the implementation of psycho-emotional stress as a trigger for the development of MACE in cardiac patients. Recently, the “smooth muscle cell contraction hypothesis” has been put forward as a cause of plaque rupture. The authors of this hypothesis proposed that MACEs arise from plaque rupture due to ischemic injury and necrosis caused by a rapid increase in CAVI in the presence of an initially elevated CAVI (Shimizu, K. et al, 2022). Our data on the association of personality type D with an increase in the START index over the course of a year confirms this possible pathogenetic mechanism.

Reviewer 2 Report

Comments and Suggestions for Authors

The manuscript has been modified. The study limitations were stated. ‘Adjusted’ hazard ratios of a decrease in the START-index for the outcome did not appear to significant. Although the confusion is made (if so, very sorry), is the conclusion supported? One more explanation may be necessary in the text for readers.

Author Response

The manuscript has been modified. The study limitations were stated. ‘Adjusted’ hazard ratios of a decrease in the START-index for the outcome did not appear to significant. Although the confusion is made (if so, very sorry), is the conclusion supported? One more explanation may be necessary in the text for readers.

Response:

We are grateful to the reviewer for this valuable comment. Indeed, multivariate Cox analysis did not reveal an independent effect of a decrease in the START index on survival and event-free survival. However, the p values ​​for survival were close to statistically significant (p=0.055). In addition, other analysis options (log-rank analysis, univariate Cox analysis) showed a significant effect of a decrease in the START index over the course of a year on survival. Therefore, we did not consider it possible to ignore these results. We recognize that baseline clinical status (eg, severity of heart failure) has a significant impact on prognosis, so the influence of risk factors (including such an integral indicator as arterial stiffness index) may not be as significant in multivariate models. Additional studies with larger numbers of participants or in more clinically homogeneous groups of patients may be needed to clarify these associations.

Reviewer 3 Report

Comments and Suggestions for Authors

The revised manuscript was finely corrected.

Comments on the Quality of English Language

No comment

Author Response

We are grateful to the reviewer for his favorable assessment of our work in correcting the text of the manuscript.

Round 3

Reviewer 1 Report

Comments and Suggestions for Authors

The authors provide adequate information to the suggestions indicated in the report. However, they should assess whether BIOMEDICINES is the appropriate journal to achieve maximum dissemination of their results and not a more specialized publication in the field of study.

Author Response

We are grateful to the reviewer for the opportunity to further evaluate our decision about the journal to publish our article. We believe that publication of our article in a special issue of “Coronary Heart Disease: Causes, Pathology and Treatment” will well contribute to achieving maximum dissemination of our results, especially given the open access policy of the journal.

Reviewer 2 Report

Comments and Suggestions for Authors

The statement of the authors is not like this reviewer does not know at all. On the other hand, as the authors acknowledged, the multivariate Cox analysis did ‘not’ reveal an independent effect of a decrease in the START index on survival and event-free survival while the p values ​​for survival were close to statistically significant (p=0.055). This reviewer was wondering if the statement is fully acceptable scientifically for most readers. Usually, the result of multivariate analysis is finally thought to be the main finding. The authors also described that differences were considered statistically significant at p<0.05 (in row 129-130 of Methods). It would be better if the further explanation that convinces anyone could be added to the Discussion. This may be a simple difference in the position of author and reviewer sides. This reviewer wants to leave the decision to the Editor.

Author Response

We are grateful to the reviewer for his comments aimed at more accurately assessing the scientific significance of the results of our study. We added the following phrase to the Limitations of the Study section: “It should also be noted that although a significant effect of a decrease in the START index over the course of a year on survival was obtained in intergroup comparisons, log-rank analysis, and univariate Cox analysis), they still do not were confirmed by multivariate Cox analysis."